3D reconstruction identifies loci linked to variation in angle of individual sorghum leaves

Tross Michael C. mtross2@huskers.unl.edu 1 2
Gaillard Mathieu 3
Zwiener Mackenzie 1
Miao Chenyong 1
Grove Ryleigh J. 1 4
Li Bosheng 3
Benes Bedrich 3 5
Schnable James C. schnable@unl.edu 1 2
1 Center for Plant Science Innovation and Department of Agronomy and Horticulture, University of Nebraska - Lincoln , Lincoln , NE , United States of America
2 Complex Biosystems Graduate Program, University of Nebraska - Lincoln , Lincoln , NE , United States of America
3 Computer Science, Purdue University , West Lafayette , IN , United States of America
4 Lincoln North Star High School , Lincoln , NE , United States of America
5 Department of Computer Graphics Technology, Purdue University , West Lafayette , IN , United States of America
Maloof Julin
Electronic publication date: 2021 Dec 22
Publication date: 2021
Volume: 9
Electronic Location ID: e12628
Received 2021 Sep 22; Accepted 2021 Nov 21
Copyright: ©2021 Tross et al.
Copyright year: 2021
Copyright holder: Tross et al.
License: This is an open access article distributed under the terms of the Creative Commons Attribution License, which permits unrestricted use, distribution, reproduction and adaptation in any medium and for any purpose provided that it is properly attributed. For attribution, the original author(s), title, publication source (PeerJ) and either DOI or URL of the article must be cited.
License URL: https://creativecommons.org/licenses/by/4.0/

Keywords: Sorghum bicolor, 3D reconstruction, High-throughput phenotyping, Leaf architecture

Funding: The Foundation for Food and Agriculture Research 602757 The USDA National Institute of Food and Agriculture and by the Office of Science (BER) 2020-68013-32371 U.S. Department of Energy DE-SC0020355 Research reported in the publication was supported by the Foundation for Food and Agriculture Research (award number—Grant ID: 602757) to Bedrich Benes and James C. Schnable. This work is supported by 2020-68013-32371 from the USDA National Institute of Food and Agriculture and by the Office of Science (BER), U.S. Department of Energy, Grant no. DE-SC0020355 to JCS. The funders had no role in study design, data collection and analysis, decision to publish, or preparation of the manuscript.

==============================
Selection for yield at high planting density has reshaped the leaf canopy of maize, improving photosynthetic productivity in high density settings. Further optimization of canopy architecture may be possible. However, measuring leaf angles, the widely studied component trait of leaf canopy architecture, by hand is a labor and time intensive process. Here, we use multiple, calibrated, 2D images to reconstruct the 3D geometry of individual sorghum plants using a voxel carving based algorithm. Automatic skeletonization and segmentation of these 3D geometries enable quantification of the angle of each leaf for each plant. The resulting measurements are both heritable and correlated with manually collected leaf angles. This automated and scaleable reconstruction approach was employed to measure leaf-by-leaf angles for a population of 366 sorghum plants at multiple time points, resulting in 971 successful reconstructions and 3,376 leaf angle measurements from individual leaves. A genome wide association study conducted using aggregated leaf angle data identified a known large effect leaf angle gene, several previously identified leaf angle QTL from a sorghum NAM population, and novel signals. Genome wide association studies conducted separately for three individual sorghum leaves identified a number of the same signals, a previously unreported signal shared across multiple leaves, and signals near the sorghum orthologs of two maize genes known to influence leaf angle. Automated measurement of individual leaves and mapping variants associated with leaf angle reduce the barriers to engineering ideal canopy architectures in sorghum and other grain crops.

Introduction

Efforts to increase yield in maize in the United States have hinged in large part, not on breeding individual plants which produce more grain, but breeding plants which can tolerate being grown more closely together, allowing farmers to grow more plants on the same amount of land (Duvick, 1984; Tollenaar, Lee et al., 2011). Retrospective phenotypic and genetic analysis has shown that breeders inadvertently selected for maize varieties with more erect leaves as they selected for plants able to grow and produce grain at high densities (Duvick, 2005). When a corn plant produces leaves parallel to the ground, the photosynthetic apparatus is often overwhelmed in high light conditions, forcing it expend energy dissipating light energy, rather than capturing all of the energy via photosynthesis (Murchie & Niyogi, 2011). On an erect leaf, the same amount of incident light is distributed across more total surface area. This increases the amount of light a plant can effectively capture and translate into stored chemical energy via photosynthesis before being overwhelmed and dissipating the excess light energy (Long et al., 2006). In maize, the classical liguleless mutants produce plants with extremely erect leaves (Moreno et al., 1997; Walsh, Waters & Freeling, 1998). Introgressing a mutant allele of the liguleless2 gene into 1960s era hybrids produced yield increases of 40% at high planting densities relative to near isogenic hybrids with the wild type allele of liguleless2 (Pendleton et al., 1968; Lambert & Johnson, 1978). Breeding for yield under high density has resulted in the leaf angle of modern maize hybrids being roughly 30°more erect (Lauer et al., 2012). More erect leaf architecture allows for denser planting, thus increasing yield per acre of land (Duvick, 2005; Ma et al., 2014)

Given its important contribution to advances in yield, the genetic control of leaf angle in grain crops has been extensively investigated through both classical and quantitative genetics in maize, as well as other grain crops. Natural allelic variation in both liguleless1 and liguleless2 contributes to variation in leaf angle among maize inbreds  (Tian et al., 2011). ZmTAC1 was first identified as a quantitative trait locus for leaf angle variation segregating in Chinese maize populations and encodes an ortholog of the rice TAC1 gene. It is primarily expressed in the leaf sheath of maize and contributes to the leaf regulatory pathway, where it is believed to be affiliated with erect angles and a more compact architecture (Ku et al., 2011). Liguleness-narrow (LGN-R) was recovered from an EMS mutant screeen and encodes a kinase upstream of liguleless1 and liguleless2 and mutant alleles produce extremely erect leaves (Moon, Candela & Hake, 2013). ZmCLA4 was mapped through fine mapping of a QTL for leaf angle and encodes an ortholog of the lazy1 gene from rice (Zhang et al., 2014). The paralogous mutants drooping leaf1 and drooping leaf2 encode YABBY transcription factors and were initially identified by classical genetics, however quantitative genetics analyses have linked these genes to natural variation in leaf angle in maize populations (Strable et al., 2017). The upright plant architecture1 (UPA1) and upright plant architecture2 (UPA2) QTL identified in a cross between maize and its wild progentior were cloned by fine mapping, revealing that upright plant architecture1 is explained by allelic variation in brassinosteroid C-6 oxidase1 and upright plant architecture2 results from allelic variation in ZmRAVL1 (Tian et al., 2019). ZmIBH1 and ZmILI1 are two basic helix-loop-helix transcription factors located on maize chromosome 2 where each appear to regulate natural variation in maize leaf angle (Cao et al., 2020; Ren et al., 2020). As of 2017, a total of approximately 12 cloned maize mutants were associated with changes in leaf angle and 19 cloned mutants in rice were associated with changes in leaf angle (Mantilla-Perez & Salas Fernandez, 2017). In sorghum, a large effect mutation that also alters plant stature, dwarf3, has been shown to alter the angle of sorghum leaves by more than 30 degrees (Truong et al., 2015).

Despite the demonstrated importance of leaf angle for increasing yields under high planting densities across a range of grain crops, investigation of the control of leaf angle variation in sorghum has been comparatively limited. Prior to the sequencing of the sorghum genome, three QTL for leaf angle were identified in a population of 137 sorghum RILs, one of which corresponded to the dwarf3 gene (Hart et al., 2001). A later QTL study employing more markers and a larger number of RILs was able to identify the effects of dwarf3 in the angle of two sequential leaves and among both field and greenhouse grown plants as well as smaller and less consistently detected QTL on chromosomes 1, 3, and 5 (Truong et al., 2015). A study of leaf angle using manually collected leaf angle scores from 315 accessions of the sorghum association panel (Casa et al., 2008) identified a significant signal from the dwarf3 locus and several additional scattered signals at the threshold of statistical significance (Zhao et al., 2016). More recently, a study which employed manual scores of leaf erectness in a 2,200 line sorghum nested association panel identified a total of sixteen significant signals scattered across the genome (Olatoye, Hu & Morris, 2020). Efforts have been made to automate the scoring of sorghum leaf angle. Leaf angle can be roughly estimated under field conditions by examining the overall width of rows of sorghum plants, a trait which can be measured from current robotic platforms (Mantilla-Perez et al., 2020). When working with individual plants in a greenhouse setting, depth camera data collected from 12 viewing angles was sufficient to reconstruct sorghum plant meshes from a panel of 99 recombinant inbred lines, enabling the detection of the known effect of the dwarf3 gene on leaf angle via QTL mapping (McCormick, Truong & Mullet, 2016). The push towards more automated approaches to measuring leaf angle in sorghum is the result of a significant bottleneck introduced by manual measurement which is labor intensive and can be error prone.

Current methods for automated measurements can track relative changes in leaf angle over time but are unable to consistently measure leaf angles across different plants with differing phyllotaxy in the case of 2D measurements (Kenchanmane Raju et al., 2020)—although it should be noted that this challenge was overcome in maize using sufficient numbers of 2D images taken from different angles (Zhang et al., 2017)—or require additional dedicated instrumentation such as depth cameras (McCormick, Truong & Mullet, 2016). Here we employ a recently published voxel carving based method for 3D reconstruction from conventional RGB photos (Gaillard et al., 2020b) and a 3D skeletonization and leaf segmentation algorithm developed to work with the same voxel data (Gaillard et al., 2020a). We extend the use of these two algorithms to allow for the measurement of the angle of each leaf of a plant and employ data on the angles of multiple leaves across multiple time points from a sorghum association panel (Casa et al., 2008) to identify both known and previously unknown genetic loci controlling natural variation in leaf angle in sorghum.

Methods

Image acquisition and annotation

A set of 366 individual plants from 236 genotypes of the sorghum association panel (Casa et al., 2008) were grown and imaged using the University of Nebraska-Lincoln’s automated phenotyping facility (Ge et al., 2016). Growth conditions and details of the image acquisition were previously described in (Gaillard et al., 2020b). RGB images were taken on April 11th (47 days after planting), 13th (49 days after planting) and 16th (52 days after planting) of 2018, using a Basler pia2400-17gc camera with a c6z1218m3-5 Pentax TV zoom lens. These images were taken at five different side view angles: 0°, 72°, 144°, 216°, and 288°side views and with a single additional image taken from directly overhead. Each image had a resolution of 2,454 × 2,056 pixels. The total number of side view images was dictated by the overall speed of the imaging system as the slowest imaging step dictated the time required to image all plants in the greenhouse. At the time, five side views was the maximum number which could be acquired by the RGB camera in the time required to capture one hyperspectral image which was the rate limiting step on data acquisition (Ge et al., 2016). Given this constraint, the angles 0°, 72°, 144°, 216°, and 288°were selected to capture five equidistant perspectives around a full 360°range of rotation. Images were calibrated as previously described (Gaillard et al., 2020b). Briefly, the calibration we applied corrects for the fact that pots are not perfectly centered on the axis of rotation of the turntable used to capture photos of the same plant from multiple angles and the axis of rotation for the turntable does not correspond to the optical center of the camera.

Two sorghum lines representing extremely different leaf architectures were selected and grown for paired manual and automated phenotyping: Btx623 (erect leaf architecture) and AS 4601 Pawaga (non-erect leaf architecure). Manual measurements of leaf angles were taken with a protractor on January 13th of 2021. Plants were imaged at three time points January 12th (70 days after planting), 13th and 14th 2021. These images were employed for 3D reconstruction and automated leaf angle measurements. Between 2018 and 2021 the physical RGB camera and lens employed by the Nebraska automated phenotyping greenhouse was replaced increasing the resolution and changing the focal length.

Plant voxelization and 3D skeletonization

A 3D volumetric grid was reconstructed for each plant on each day using the five side view images from different angles and single top view image described above and an extended voxel carving algorithm (Gaillard et al., 2020b). The algorithm employed segmented images from each view to carve a voxel grid with a resolution of 5123 voxels. The voxels remaining after carving were used to generate a 3D skeleton of the plant (Gaillard et al., 2020a).

Briefly, the generic voxel thinning algorithm implemented in the DGTal library was first run on the voxel set V. This algorithm iteratively removes voxels from the reconstructed plant V until only its curve skeleton Sraw remains. This curve skeleton is a collection of curves whose shape faithfully represents the reconstructed plant. However, the curve skeleton Sraw includes some small spurious branches that do not correspond to separate organs of the plant being reconstructed. A machine learning classifier was run on the curve skeleton Sraw to distinguish between spurious branches and branches that correspond to real leaves with the spurious branches being discarded (Gaillard et al., 2020a). It is possible that the voxel grid contains more than one connected component i.e., the reconstructed plant is disconnected. In this case, the skeletonization algorithm will automatically connect branches together to form the plant skeleton. We denoted the final plant skeleton S. Finally, the skeleton S is post-processed to identify the stem T and each individual leaf L=L1,L2,…,Ln. To differentiate voxels that are part of the stem or individual leaves, we consider that 1) any voxel that is shared by at least two branches is part of the stem, and 2) any voxel that is part of only one branch belong to the corresponding leaf (Gaillard et al., 2020a). Leaves were numbered based on the height of each leaf’s leaf-stem junction with the first leave being the leaf which joins the stem at the lowest point and the last leaf being the detected leaf which joins the stem at the highest point.

Leaf angle measurement from 3D skeletons

The angle of individual leaves is measured in a three step process. In the first step, the principal direction v1,T → and the second direction v2,T → of the stem are identified and measured by analyzing the 3D coordinates of voxels assigned to the stem of the plant skeleton T using PCA (Principal Component Analysis). In the second step, the principal direction vLn → of the first 20 voxels of each leaf (the 20 voxels after the leaf-stem junction) Ln are computed, also through the use of the principal component analysis. In the final step, the principal directions of the stem and leaf are compared to compute two angles: (1) The polar angle θ ∈ [0°, 180°], which is the angle in the vertical plane, and (2) the azimutal angle ϕ ∈ [0°, 360°], which is the angle in the plane formed by v1,T → and v2,T →. The angle θ is computed as the angle between the two 3-D vectors vLn → and v1,T →. The angle ϕ is computed as the angle between v2,T → and the projection of the leaf direction in the horizontal plane vLn →−vLn →⋅v1,T →v1,T →.

Leaf angle measurements from 2D images

For each plant at each time point, all captured viewing angles were segmented into plant and not-plant using the published PlantCV protocol (Fahlgren et al., 2015; Gehan et al., 2017). Briefly, images were converted to grayscale, blurred, cropped to remove pots and backgrounds. These gray scale images were then converted to binary segmented images using the PlantCV thresholding function. Plant width was calculated as the distance between the left-most and right-most plant pixels in each image, and for each plant on each day the viewing angle with the greatest plant width was selected as the one most likely to represent a plane of phyllotaxy roughly perpendicular to the viewing angle of the camera. This widest image was skeletonized, the skeleton divided into leaf segments and the insertion angles of each leaf segment were calculated following the “morphology” protocol workflow from PlantCV’s documentation (Gehan et al., 2017).

Heritability and genome-wide association

Broad sense heritability was calculated using 48 replicated genotypes included in the study using the equation H2=σG2/σG2+σe2/2, where σG2 is the total amount of variance that is due to genetics and σe2 is the total residual variance. The variance components were extracted from the linear model: yi = μ + ti + ei, where yi is the mean yield of the genotype, µis the overall mean, ti is the effect of genotype i and ei is the residual error of genotype i. This was implemented in the software package lme4 (v1.1-23)(Bates et al., 2015).

Best linear unbiased predictors (BLUPs)(Robinson et al., 1991) were calculated for each set of leaf angle measurements used for GWAS with leaf angle treated as a random effect using the same linear model: yi = μ + ti + ei. Any genotype with a BLUP more than 5 standard deviations away from the mean was treated as an outlier and removed from downstream analysis. Each outlier value was checked against image data to confirm the extreme values resulted from the plant itself and not reconstruction errors. In total one genotype was removed as an outlier for leaf 1, 1 was removed as an outlier for leaf 2, and one was removed from the aggregated leaf angle across leaves 1–4.

The genetic marker dataset employed in this study consisted of 569,306 SNP markers published as part of Miao et al. (2020a). The dataset was filtered to remove markers with a minor allele frequency of <0.05 and a heterozygousity of >0.05 among the 236 genotypes phenotyped in this study. This produced a set of 232,113 SNP markers employed for GWAS. GWAS was conducted using both mixed linear model (Yu, Pressoir & Briggs, 2006) and FarmCPU GWAS algorithms. MLM based GWAS was conducted using the GEMMA software package (v0.98.1) (Zhou & Stephens, 2012). FarmCPU based GWAS was conducted using the algorithm as implemented in the rMVP software package (v1.0.2) (Yin et al., 2020). In each model three principal components calculated from genetic marker data were incorporated as covariates. In addition, for the analysis conducted using GEMMA, a kinship matrix was fit as a random effect. For GEMMA, a statistical significance threshold of 6.39 × 10−7 equivalent to 0.05/78251 SNPs was employed. The number of SNPs used in calculating this statistical significance threshold was the effective number of independent tests represented in this genetic marker dataset calculated using GEC v.02 (Li et al., 2012). For FarmCPU, a subsampling strategy was employed to identify stable genetic associations. 100 random subsets of 87% of the data equivalent to 207/236 genotypes, were generated, and each was analyzed separately using the FarmCPU algorithm as described above. Individual SNPs which were assigned p-values more significant than 2.15 x 10−7 equivalent to 0.05/232,113 SNPs were treated as positive identifications. SNPs which were positively identified as linked to leaf angle in at least 10 of the 100 resampling analyses were classified as significant and stable associations.

Results

Measurements from 3D reconstructions of sorghum plants

Initially two lines representing phenotypic extremes for leaf angle in the sorghum association population were selected to evaluate leaf angle measurement: BTx623, the reference genotype for sorghum which produces erect leaves (Fig. S1A), and AS 4601 Pawaga which produces non-erect leaves (Fig. S1B). Six plants were grown to late vegetative stage, the angles of the first four non-senesced leaves were manually measured, and the plants were imaged using a previously described automated greenhouse imaging system (Ge et al., 2016). Sets of 2D images collected from each plant from multiple viewing angles were processed to generate 3D voxel reconstructions using the method described in Gaillard et al. (2020a) (Fig. 1A). Three dimensional skeletonization and pruning was employed to identify individual leaves (Gaillard et al., 2020a) (Fig. 1B). And the angle of individual leaves relative to the principle direction of the stem was quantified from these segmented skeletonized leaves (Fig. 1C). The manual and Automated 3D measurements of median leaf angles across the first four leaves were highly positively correlated r = 0.98 with manual measurements taken with a digital caliper (Figs. S3A; S2). Manual and automated measurements for individual leaves exhibited a modestly lower but still high correlation r = 0.86 (Fig. S3B).

Figure 1 Measurement of individual leaf angles from 3D reconstructions of sorghum plants.

(Aa) An example of one of the six view 2D images—five side views and one top view—used to reconstruct the 3D volume of the plant using voxel carving. (Ab) Evaluation of the quality of the voxel reconstruction by comparison the initial segmentation of this 2D image for plant and not-plant pixels to a reprojection of the voxel reconstruction as a 2D image viewed from the same perspective. Green pixels mark overlap between these two images. Red pixels are places identified as part of the plant in the original segmented image but not the reprojection. Blue pixels are places which are part of the projection but were not identified as part of the plant in the original 2D segmentation. (Ac) A 3D skeleton (black lines) fit to the voxel reconstruction. (Ad) Overlay of the 3D skeleton on the original RGB image from (Aa). (B) Measurement of individual sorghum leaf angles in 3D space. Separate vectors are generated for the stem (blue) and leaf (yellow) using the voxel-based skeletons of each organ. Leaf angle was defined as the polar angle θ (the angle with regard to the stem principal direction). These measurements also reconstructed a second azimutal angle ϕ (angle in the plane formed by v1,T → and v2,T →). (C) The plant skeleton with measured angles for each leaf indicated. The solid red line indicates the stem principal direction while black lines mark the principal directions for each leaf.

Having evaluated the performance of this 3D approach to leaf angle measurement on a small set of plants with ground truth manual measurement, the algorithm was next applied to a much larger experiment from which ground truth measurements had not been collected: An image dataset collected at multiple time points from a total of 366 sorghum plants representing 236 unique genotypes were used to create 1092 3D reconstructions, each representing a single plant on a single day. A total of 119 3D reconstructions failed. In 39 cases corresponding to 13 pots on each of the three time points no plant was present in any image, presumably the result of failed germination or the inclusion of calibration pots. In 35 cases calibration was unsuccessful. In the remaining 45 cases either the concordance between the original segmented 2D images and simulated 2D segmented images generated from 3D reconstructions was below 0.8 (Gaillard et al., 2020b), or the 3D skeletonization algorithm was not able to confidently identify the soil-stem junction (Gaillard et al., 2020a). Of the remaining 973 3D reconstructions, 2 were removed based on unexpectedly extreme leaf angle values. Manual evaluation of these two outlier reconstructions suggested that one resulted from a small plant with a large tiller (Fig. S4A). The reason for the poor reconstruction of the second outlier plant was not immediately apparent (Fig. S4B).

Sorghum plant reconstructions for the remaining 971 unique plant/timepoint combinations contained between 3 and 19 predicted leaves, with a median of 10 (Fig. S5). Median leaf angle declined, e.g., became more erect, starting from the bottom most leaf and proceeding upwards to leaf six (Fig. S6). Leaf angles measured from 3D skeletons for leaves above leaf six exhibited a much wider distribution of values, which may reflect the difficulty of accurately quantifying leaf angle for leaves still emerging from the leaf whorl where the ligule/auricle junction is not yet visible. It was not possible to automate the separation of mature and emerging leaves, so it was unclear how much accuracy was lost for uppermost leaves.

As this analysis was conducted using a previously collected sorghum image dataset for which ground truth leaf angles were not manually scored, it was not possible to directly evaluate how well this larger set of measured leaf angles represented the angles which would have been measured by hand from the exact same plants. However, because the genotypes used in this study were largely homozygous inbred lines, it was possible to grow plants with identical or near identical genomes to the same stage of development and collect ground truth measurements from these plants. Independent grow outs of the same set of genotypes do not produce perfectly equivalent phenotypes. Measurements of plant leaf angle collected in Iowa and Nebraska from the same population were compared in order to estimate how much correlation in leaf angle would be expected between two independently grown sets of plants at the same stage of development—reproductive stage in this case—collected in different environments. The correlation between these two manual datasets data was significant (r = 0.60; p = 0.0002; pearson). Leaf angle measurement were manually scored from a set of 79 lines from the sorghum association panel grown to a similar late vegetative stage of development under greenhouse conditions were significantly correlated with automated 3D measurements (r = 0.53; p = 0.0016; pearson) (Fig. S7). However, analysis of the same image data using conventional 2D approaches to quantifying leaf angle from single images produced values which were not significantly correlated with either manual measures of the same genotypes at the same stage, or leaf angles measured from 3D reconstructions using the same image data (Fig. S7). Manual measurements of plant leaf angle between Iowa and Nebraska datasets were anticipated to be higher than the automatic 3D versus manual measurements of the greenhouse plants where different plants of the same genotypes were compared. The leaf angles for each genotype in both the Iowa datasets and Nebraska dataset of mature plants were the means of up to 18 and 2 replicates of genotypes respectively. This would reduce environmental, genotype by environment and many other residual effects. The dataset composing of 3D reconstructed plants and that of the 79 lines under greenhouse conditions generally represented measurements of multiple leaves from only a single plant.

Assuming random error in quantification, as opposed to genetically controlled error (Liang et al., 2017), if the angles of upper leaves are measured with lower accuracy, the heritability—i.e., the proportion of total variance attributable to differences between genotypes—of leaf angle measurements for these leaves would likely be lower. A total of 48 sorghum genotypes were replicated between two and eight times each among the 366 unique sorghum plants imaged. Only the first seven leaves of each plant were considered in our analysis. Higher leaves were generally not mature as evident by the auricle not being extended from the sheath. This would limit the accuracy of leaf angle measurements of those after the seventh leaf. The heritability of 3D leaf angle measurements tended to be higher for the first five leaves than for leaves six or seven on each of the three time points measured (Fig. S8A). The median measurement for the angle of a particular leaf across three time points tended to be more heritable than measurements of the same leaf at individual time points, consistent with expectations for repeated measurements with independent error (Fig. S8A). Aggregating measurements across sequential leaves also tended to increase heritability relative to single leaf measurements (Fig. S8B). Of the aggregated leaf ranges within the aggregated time points,the median of leaves 1 to 4 had the highest heritability value of 0.72 (Fig. S8B). The heritability of measurements of leaf angle obtained from conventional 2D analysis of the same set of sorghum images were generally much lower than 3D measurements with no genetic effect detected in a number of cases (Fig. S9).

Genetic loci associated with leaf angle variation in the SAP

A published set of SNP markers for the sorghum association panel (Miao et al., 2020a) was employed to conduct a genome wide association study for both the 3D and 2D aggregated leaf angle data across the first four leaves and all three time points. Analysis with GEMMA (Zhou & Stephens, 2012), a mixed linear model based algorithm, did not identify any significant markers for the automated 2D angles (Fig. S10) but for 3D derived angles, identified a single large peak for leaf angle located on chromosome 7 between 59.8 and 59.9 MB. This position corresponds to the genomic location of dwarf3 (Sobic.007G163800), an ATP-binding cassette (ABC) transporter involved in auxin export (Multani et al., 2003) that has been previously associated with variation in leaf angle in sorghum RIL populations (Truong et al., 2015) and an association study (Mantilla-Perez et al., 2020). As dwarf3 is a large effect locus with functionally variable alleles segregating at high frequencies in this population, we speculated that the effects of additional loci on angle might be masked in the MLM-based GWAS results (Fig. 2A). The FarmCPU algorithm, a multi-locus GWAS algorithm that can control for large effect loci to increase the likelihood of identifying additional true positive signals, was employed to analyze the same trait and genotype datasets. The significance of FarmCPU results were controlled for using a resampling-based strategy (Valdar et al., 2009). A total of eight markers clustered at six locations in the genome showed significant associations (resample model inclusion probability (RMIP) > 0.1) with median angle of leaves 1–4 across all three time points in > 10% of resampled FarmCPU GWAS results, including the dwarf3 locus (Fig. 2B). None of the five non-dwarf3 clusters were associated with a set of 19 a priori candidate sorghum genes identified based on syntenic orthology to genes known to influence leaf angle in maize or rice. However, three of the six clusters associated with variation in leaf angle were consistent with QTL detected in a nested association mapping study for leaf erectness using a population of 2,200 sorghum RILs (Olatoye, Hu & Morris, 2020).

Figure 2 Genetic markers significantly associated with variation in sorghum leaf angle aggregiated across leaves and time points.

(A) Results of a genome wide association study conducted using the GEMMA algorithm. Each point indicates the physical position and statistical significance of an individual marker. Dashed black line indicates a genome wide threshold for statistical significance of 6.39 × 10−7 resulting from a bonferroni correction using an effective SNP number of 78,251 (See Methods). (B) Results of a resampling based assessment of associations identified using the FarmCPU GWAS algorithm. Each circle represents an individual genetic marker which was statistically significantly associated with leaf angle variation in at least one of 100 FarmCPU analysis conducted using subsets of the total phenotypic dataset. x-axis position indicates the physical position of the marker and y-axis position indicates the proportion of the 100 analysis in which the marker was identified as statistically significant. Dashed black line indicates a threshold cutoff for stable and significant associations which were detected in >10% of total resampling analyses. Yellow triangles indicate the locations of significant QTL for leaf erectness detected in an analysis of a sorghum NAM population (Olatoye, Hu & Morris, 2020). Black triangles indicate locations of a set of cloned sorghum genes or the locations of the syntenic ortholog in sorghum of maize or rice genes known to influence variation in leaf angle in those species..

Figure 3 Genetic markers significantly associated with variation in the angle of leaves 1, 2, or 3.

A resampling based assessment of associations identified using the FarmCPU GWAS algorithm using the median leaf angle for leaf 1, 2, or 3 across all three time points. As in Fig. 2, the dashed black line indicates a threshold cutoff for stable and significant associations which were detected in >10% of total resampling analyses. Yellow triangles indicate the locations of significant QTL for leaf erectness detected in an analysis of a sorghum NAM population (Olatoye, Hu & Morris, 2020). Black triangles indicate locations of a set of cloned sorghum genes or the locations of the syntenic ortholog in sorghum of maize genes which are both known to influence variation in leaf angle in maize and are located near significantly trait associated SNPs in this study..

The reconstruction of full architectures of individual plants provided the opportunity to assess how much overlap was present in the loci showing significant associations with variation in leaf angle for individual leaves. Separate analyses were conducted for the angle of the first, second, third, and fourth identified leaves of each sorghum plant. It must be noted that these likely do not correspond to the leaves which would be called leaf 1, leaf 2, leaf 3, and 4 as early sorghum seedling leaves senesce and die as the plant continues to develop. However, the first identified leaf should always be an earlier developing leaf than the second and third and and so on. Eight total markers showed significant associations in >10% of sub-sampled GWAS results for the angle of the first identified leaf although three of these were grouped together in the pericentromeric region of chromosome 3 and may represent a single underlying signal, five clusters of ≥ 1 SNPs showed significant associations in >10% of sub-sampled GWAS results for the second identified leaf, five showed significant associations in >10% of sub-sampled GWAS results for the third identified leaf (Fig. 3), and no significant signals were identified for the fourth identified leaf. The signal associated with the dwarf3 locus was detectable for the second and third leaves but not the first leaf. One of the six significant clusters identified in GWAS for the angle of the first leaf was located on sorghum chromosome 2, 420 kilobases from the sorghum ortholog (Sobic.002G272400) of ZmTac1 (Ku et al., 2011). One of the five significant associations for the second leaf was located at 54.99MB, 503 kilobases from the sorghum ortholog (Sobic.006G190400) of the maize gene UPA2/ZmRAVL1 (Tian et al., 2019). In addition to the dwarf3 locus, a second shared signal was identified between GWAS for the angle of the second identified leaf and GWAS for the angle of the third identified leaf at position 52.75MB on sorghum chromosome 10. This hit did not correspond to either published loci associated with leaf erectness in the sorghum NAM population or sorghum orthologs of candidate leaf angle regulatory genes from maize or rice. Mixed linear model based, leaf by leaf GWAS results, showed 3 significant associations for leaf 2 and one for leaf 3 (Fig. S11). Two of the associations for leaf 2, one on chromosome 7 and the other on 10 was in close proximity to signifcant SNPs via the leaf by leaf FarmCPU resampling (5.031 and 29.997 kilobase differences respectively). The association for leaf 3, on chromosome 6, was also in close proximity to a significant SNP found in the FarmCPU resampling method (34.572 kilobase difference).

Discussion

Leaf angle has been shown to play a role in determining photosynthetic productivity of grain crops at differing planting densities. In sorghum, quantitative genetic investigation of the genes controlling natural variation in leaf angle has been limited by the labor intensive nature of manual leaf angle measurements. Here we employed voxel-based 3D reconstruction of greenhouse grown plants combined with 3D skeletonization to quantify leaf angles for each individual leaf of 366 sorghum plants at three time points, representing 971 individual 3D reconstructions—after removing failed reconstructions—and 3,376 total individual leaves. We chose to focus on the angle between leaves and stems which more directly corresponds to manual measurements used by plant geneticists to identify genes controlling variation in leaf angle (Mantilla-Perez & Salas Fernandez, 2017) (Fig. S2). Automated measurements of the angle of lower leaves from 3D reconstructions—which tend to be fully expanded and ligulated—were positively correlated with manual measurements of the same plants in a small study (Fig. S3), correlated with manual measurements of plants of the same genotype in a larger study (Fig. S7), and heritable in a study employing the sorghum association population (Fig. S8A).

Aggregated leaf angle data extracted from 3D reconstructions was sufficient to identify dwarf3 (Fig. 2A), a locus with known effects on leaf angle in sorghum (Truong et al., 2015; McCormick, Truong & Mullet, 2016). This known leaf angle gene could not be identified using leaf angle measured obtained from applying conventional 2D approaches to measuring leaf angles to the same image dataset. Analysis of the same dataset with the FarmCPU GWAS algorithm identified five additional signals, two of which were consistent with previously reported leaf angle QTL (Olatoye, Hu & Morris, 2020). However, while the majority of previous quantitative genetic analyses of leaf angle in sorghum focused on either a single leaf or an average across leaves (Mantilla-Perez & Salas Fernandez, 2017), our 3D reconstructions made it possible to conduct separate analyses for variants influencing the angle of the four lowest non-senesced leaves. Significant signals were identified for three of these four leaves including a known leaf angle variant from sorghum, signals near the sorghum orthologs of maize genes known to influence leaf angle, and a single independently identified for the angle of leaves two and three which was not linked to previous QTL in sorghum nor near orthologs of known leaf angle related genes in relatives (Fig. 3). However, many loci showing significant associations with the angle of only one of the leaves tested.

It would be tempting to interpret this as evidence of largely independent genetic architectures controlling the angle of each individual sorghum leaf. More pessimistically the same result could be interpreted as many of the identified associations representing false positive associations. There is indeed evidence that genetic variants associated with upper and mid-level leaf angle in sorghum are only partially overlapping (Mantilla-Perez et al., 2020). However, in this case we examined the angle of three sequential leaves rather than distinct parts of the sorghum canopy. In addition, as a result of different rates of senescence for juvenile leaves, there is no guarantee that the “leaf one” of two distinct sorghum genotypes in this study was a biologically equivalent leaf. Should we then conclude that the low degree of overlap between the trait associated SNPs for the angle leaves one, two and three results from a high rate of false positives? Not necessarily. For traits controlled by complex genetic architecture, such as leaf angle in sorghum, statistical approaches to GWAS must accept a high degree of false negatives in order to minimize false positives. In simulation studies using the sorghum association panel, for traits with the observed heritability of individual sorghum leaves controlled by as few as 64 total genetic variants, less than 40% of total variants are predicted to be discovered by the FarmCPU algorithm –and even less using more conventional MLM based approaches (Miao, Yang & Schnable, 2019). The simplest interpretation of the low amount of overlap between the significant signals identified for the angle leaves 1, 2, and 3 may that each represents a different subset of the true variant associations with lower leaf angle in sorghum, identified from independent datasets which all reflect a mixture of consistent genotype effects, genes with increasing or decreasing effects from lower to upper leaves, leaf by leaf non-genetic variation, and measurement error.

This study demonstrated that aggregating measurements of the same plant across multiple time points, even time points separated by only several days, can increase the proportion of variance explained by inter-genotype variation. By imaging in a controlled environment with artificial lighting, we were able to maintain largely constant lighting conditions across time points. Changes in illumination in field or greenhouse environments would make obtaining accurate segmentation across time points more challenging (Adams et al., 2020). However, assuming accurate segmentation of 2D images, changing in lighting conditions would not alter the accuracy of 3D reconstruction based on the voxel carving approach employed here. To more accurately leverage data from multiple time points as well as reliably track changes in leaf angle over time in response to environmental changes, it would be necessary to track each leaf across successive reconstructions of the same individual plant. This is a computationally challenging for at least three reasons. Firstly, successive reconstructions of the same plant may not be aligned as a result of affine transformations resulting from rotation of the plant and pot. Secondly, leaves themselves grow and change over time. Individual leaves emerge from the whorl, lengthen and lower their observed angle with respect to the stem until the leaf reaches maturity. Later the lowest leaves will tend to senesce, wither, and die to the point they will no longer be detectable on subsequent 3D reconstructions. Thirdly, discrete events can occur which significantly alter the architecture of the plant, particularly the emergence of a newly detectable leaf from the whorl, or the senescence, withering, and death of lower leaves to the point they will no longer be detectable on subsequent 3D reconstructions.

There are conceptually straightforward computational approaches which could be adopted to address the first two challenges. Considering two successive reconstructions Pt and Pt+1 of the same plant P at time points t and t + 1, it is possible to find the optimal homography to align Pt and Pt+1, minimizing the distance between them. Subsequently, it is possible to measure the distances between all pairs of leaves from Pt to Pt+1, and compute a minimum matching between them (for example by using the Hungarian algorithm (Kuhn, 1955; Edmonds & Karp, 1972)). This would give us for each leaf of Pt, the corresponding leaf in Pt+1, while minimizing the total distance between all pairs of matched leaves. By successively tracking leaves from P1 to P2, then P2 to P3, all the way up to Pn, we track each leaf over the entire life span of the plant. Unfortunately, this method has one major drawback, it assumes the number and identity of leaves is constant between data collected from the same plant at different time points, causing it to fail when confronted with the third challenge outlined above: the emergence of new leaves or the senescence and death of old ones. Further work is needed to extend leaf tracking across multiple time points to take into account these discrete events which change the number and/or identity of detectable leaves present.

In conclusion, here we demonstrated the ability to quantify individual leaf level angle measurements in sorghum using 3D reconstructions. The method does not require specialized equipment such as depth cameras or LIDAR sensors and can scale to populations of hundreds of plants. We demonstrate that leaf angle measurements collected in this fashion can be used to identify both known and novel loci in the sorghum genome influencing overall leaf angle or the angle of individual sorghum leaves. 3D reconstruction and leaf angle measurement—as well as the measurement of other leaf properties including length, width, and curvature—across multiple stages of development can enable a more comprehensive understanding of the genetic determinants of sorghum canopy architecture and aid breeding or engineering of more productive and resource use efficient “smart canopies” (Mantilla-Perez et al., 2020).

Supplemental Information

Supplemental Information 1 Two genotypes having different extremes of leaf angles were selected to compare manual and automatic measurements of the same plants

(A) BTx623 (B) AS 4601 Pawaga.

Click here for additional data file.

Supplemental Information 2 Example of manual leaf angle measurement in sorghum

An electronic protractor (Husky Sliding Digital T-Bevel/Angle Finder) is used to measure the interior angle between the sorghum stem and the midrib of the target sorghum leaf.

Click here for additional data file.

Supplemental Information 3 Automatically derived leaf angle measurements from 3D reconstructions are positively correlated with manual measurements of the same plants

(A) Pearson r correlations between the median leaf angle of leaves 1 to 4 derived from manual measurements of six sorghum plants representing phenotypic extremes for leaf angle in sorghum versus the median leaf angle of the same plants derived from automatic measurements of the 3D reconstructed plants. (B) Pearson r correlations between manual measurements of the individual leaves 1 to 3 of six sorghum plants representing phenotypic extremes of leaf angle in sorghum versus automatic measurements of the same plants derived from 3D reconstructions.

Click here for additional data file.

Supplemental Information 4 Overlay of outlier plants removed from all analyses and their optimized skeletons

(A) Leaf angles were large due to biological errors as a result of extremely poor growth and health under greenhouse conditions. (B) Leaf angles for plant was inconsistent with visual validation.

Click here for additional data file.

Supplemental Information 5 Frequency of the total amount of leaves of each plant

Click here for additional data file.

Supplemental Information 6 Violin plots illustrating the range of leaf angle values for each leaf

Click here for additional data file.

Supplemental Information 7 Pearson correlations and histograms of pairwise combinations of leaf angle datasets (∗∗∗=p < 0.001, ∗∗=p < 0.01, ∗=p < 0.05)

Manual Field 1, Manual Field 2, Automated 3D and Automated 2D refers to the Iowa 2012 manual field data, the Nebraska 2020 manual field data, the Nebraska 2021 manual green house data, the Nebraska 2018 greenhouse automatically derived 3D data and the Nebraska 2018 greenhouse automatically derived 2D data. Stronger correlations were observed between datasets grown under the same condition and maturity stage (Manual Field1 vs Manual Field2 and Manual greenhouse vs Automated 3D). Negative and non significant correlations observed between comparisons of automatically derived 2D measurements and all other datasets.

Click here for additional data file.

Supplemental Information 8 Heritability increases with the aggregation across time points and leaves within a plant

(A) Heritability calculated for each individual leaf for each of the three time points as well as after aggregation of time points. (B) Heritability calculated using the median value of different leaf combinations for three different time points and after aggregation (The median angle of the three values of a single leaf, was assigned for that specific leaf and plant).

Click here for additional data file.

Supplemental Information 9 Lower heritability values generally observed in leaf angles obtained from automated 2D measurements

(A) Heritability calculated for each individual leaf for each of the three time points as well as after aggregation of time points. (B) Heritability calculated using the median value of different leaf combinations for three different time points and after aggregation (The median angle of the three values of a single leaf, was assigned for that specific leaf and plant).

Click here for additional data file.

Supplemental Information 10 No genetic markers significantly associated with variation in sorghum leaf angle aggregated across leaves and time points with automatic 2D derived measurements

A Mixed linear model based approach implemented in GEMMA [39] was used to identify significant associations of SNP markers with variation in leaf angle aggregated across all three time points. Each point indicates the physical position and statistical significance of an individual marker. Dashed black lines indicate a genome wide threshold for statistical significance of a 6.39 × 10 −7 resulting from a bonferroni correction using an effective SNP number of 78,251 (See Methods). Yellow triangles indicate the locations of significant QTL for leaf erectness detected in an analysis of a sorghum NAM population (Olatoye, Hu & Morris, 2020. Black triangles indicate locations of a set of cloned sorghum genes or the locations of the syntenic ortholog in sorghum of maize genes which are known to influence variation in leaf angle in maize and are located near significantly trait associated SNPs in this study.

Click here for additional data file.

Supplemental Information 11 Genetic markers significantly associated with variation in the angle of leaves 1, 2 and 3

A Mixed linear model based approach implemented in GEMMA [39] was used to identify significant associations of SNP markers with variation in leaf angle aggregated across all three time points. Each point indicates the physical position and statistical significance of an individual marker. Dashed black lines indicate a genome wide threshold for statistical significance of a 6.39 × 10 −7 resulting from a bonferroni correction using an effective SNP number of 78,251 (See Methods). Yellow triangles indicate the locations of significant QTL for leaf erectness detected in an analysis of a sorghum NAM population (Olatoye, Hu & Morris, 2020. Black triangles indicate locations of a set of cloned sorghum genes or the locations of the syntenic ortholog in sorghum of maize genes which are known to influence variation in leaf angle in maize and are located near significantly trait associated SNPs in this study.

Click here for additional data file.

Additional Information and Declarations

Competing Interests

Author Contributions

Data Availability

James C. Schnable has an equity interests in Data2Bio, a company that provides genotyping services using the same protocol employed for genotyping in this study.

Michael C. Tross, Mathieu Gaillard and Ryleigh J. Grove performed the experiments, analyzed the data, prepared figures and/or tables, authored or reviewed drafts of the paper, and approved the final draft.

Mackenzie Zwiener and Chenyong Miao performed the experiments, authored or reviewed drafts of the paper, and approved the final draft.

Bosheng Li performed the experiments, analyzed the data, authored or reviewed drafts of the paper, and approved the final draft.

Bedrich Benes and James C. Schnable conceived and designed the experiments, performed the experiments, authored or reviewed drafts of the paper, and approved the final draft.

The following information was supplied regarding data availability:

The code for reconstruction and skeletonization is available at GitHub: https://github.com/cropsinsilico/SorghumVoxelCarving.

The raw images analyzed in this study are available at Zenodo: Mathieu Gaillard, Chenyong Miao, James C. Schnable, & Bedrich Benes. (2021). Voxel Carving Based 3D Reconstruction of Sorghum [Data set]. Zenodo. https://doi.org/10.5281/zenodo.4426620.

The phenotypic data, GWAS result files and code for main figures are available at GitHub: https://github.com/mtross2/Sorghum-3D-Reconstruction.

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
