# Peer review of "D reconstruction identifies loci linked to variation in angle of individual sorghum leaves"

_PeerJ, doi:10.7717/peerj.12628_

## Round 0.1 · original submission · Minor Revisions

Thank you for submitting this work to PeerJ. Both reviewers are enthusiastic about this manuscript but request some minor revisions. Please submit a revised version that addresses their concerns.

·

Basic reporting

This research proposed an automated measurement of individual leaves and mapping variants associated with leaf angle. The reconstruction of 3D geometry of individual sorghum plants using a voxel carving method was effective, the computation of leaf angles based on curve skeleton of the 3D geometry was reasonable. Experiment design and results are reasonable and statistically sound. Some details need to be improved as follows:

Experimental design

1. Line 92, please explain the reason and theory support to choose five different side view angles: 0, 72, 144, 216, and 288 as side views.
2. Line 96, since the imaged at three time points, does the illumination change with different time points? Does the illumination condition will affect the reconstruction of 3D voxels?
3. Line 101, how many images were used for each plant to reconstruct 3D volumetric grid? Please add this detail into the manuscript.
4. BTW, there might be a typo, “volumentric” should be change to “volumetric”.
5. Line 111, is there any disconnection part in the voxel reconstruction model?
6. Line 122, the definition of leaf angle was based on the principal direction of stem and leaf direction, since the principal direction of stem will change with different plants, it is a relative local angle. It would be better to convert them to an angle between leaf direction and Z axis in the same 3D coordinated system. So, the comparison of leaf angles among different plants will be reasonable and easy.
7. Line 160, Figure 1, “B) Measurements The plant skeleton represented with measured angles for each leaf. The solid red line indicates the stem principal direction. Solid black lines represent the leaf principal directions.” Please change the description of B). The solid red line and Solid black lines should be a description in C) instead of B).
8. Line 169, how to measure the median leaf angles and the related principal direction of stem in a manuall manner? What instrument was used? Is it possible to add one Figure to illustrate it?
9. Line 176, what is the main reason causing the 119 reconstructions failed?
10. There is no detail explaining the “calibrated” 2D images mentioned in the abstract. Suggest adding this into Methods part.

Validity of the findings

Experiment design and results are reasonable and statistically sound.

·

Basic reporting

Summary
* * *
In the manuscript “3D reconstruction identifies loci linked to variation in angle of individual sorghum leaves,” Tross and colleagues utilize a method for 3D reconstruction from multiple 2D images with a sorghum dataset from an automated greenhouse facility. The 3D reconstructions of the sorghum plants are converted to a node-edge graph structure, which are used to separate the sorghum plants into stem and leaf segments. In 3D space, the segmented plant structures were used to measure the angle between leaves and the stem to assess leaf erectness, which is a difficult trait to measure by hand. Automated 3D measurements of leaf angle had high correlation with manual measurements, even when considering measurements collected on different sets of plants in different environments. These measurements outperformed automated measurements from 2D images alone. Automated 3D leaf angle measurements were shown to be generally highly heritable and in GWAS analyses were used to identify an association peak containing the previously identified dwarf3 locus in addition to five other loci.

Strengths
* * *
- The manuscript is well-written, clear, and concise.
- The computational approach is interesting and addresses an important problem in plant phenotyping and crop development.
- The methods are complete and data and code are available for replication and reuse.

Weaknesses
* * *
None of note.

Checks
* * *
Clear and unambiguous, professional English used throughout: Yes (minor grammatical suggestions below).

Literature references, sufficient field background/context provided: Yes, the introduction was excellent and relevant literature was cited.

Professional article structure, figures, tables. Raw data shared: Yes

Self-contained with relevant results to hypotheses: Yes

Experimental design

Checks
* * *
Original primary research within Aims and Scope of the journal: Yes

Research question well defined, relevant & meaningful. It is stated how research fills an identified knowledge gap: Yes

Rigorous investigation performed to a high technical & ethical standard: Yes

Methods described with sufficient detail & information to replicate: Yes, data and analysis code are shared for reuse and reproducibility. The methods in the paper are well-described and include the versions used.

Validity of the findings

Checks
* * *
All underlying data have been provided; they are robust, statistically sound, & controlled: Yes, data are provided and statistical approaches are sound and documented.

Conclusions are well stated, linked to original research question & limited to supporting results: Yes, conclusions are well supported by the data.

Additional comments

Minor Suggestions
* * *
- It is ideal to attach a license to your GitHub repository (https://github.com/mtross2/Sorghum-3D-Reconstruction) to define the conditions for use, modification, redistribution, and attribution (as applicable).
- Figure 1B (legend): Description of the colors of the lines do not match the figure.
- The quality of Supplemental Figures S2, S4, and S5 look like they were degraded by conversion to PDF. Likely just need to make sure the uploaded file type or resolution works well with the PeerJ system, the figure quality is good otherwise.

Minor grammatical suggestions
* * *
- Line 59: “In sorghum, ...”
- Line 174: “An image dataset…” or “Image data…”
- Lines 175-176: Could you succinctly add why 119 timepoints failed reconstruction?
- Lines 208: “Only the first seven leaves of each plant were…”
- Line 225: “...that has been previously associated…”
- Line 227: “...the effects of additional loci on angle...”
- Line 228: “...GWAS algorithm that can...”
- Line 251: italicize dwarf3
- Line 252: “...third identified leaf at position...”
- Line 260: “Leaf angle has been shown to play a role in determining...”
- Line 293: “...true variant associations…”
- Line 309: “There are conceptually…”

---

## Round 0.2 · accepted · Accept

Thank you for carefully addressing the reviewer's points, and thank you for submitting this interesting work to PeerJ. I am recommending this for acceptance.